# Analysis of generalization capacities of Neural Ordinary Differential Equations

**Madhusudan Verma**[*]                                                   *zda23m015@iitmz.ac.in*
*School of Engineering and Science*
*Indian Institute of Technology Madras, Zanzibar Campus*

**Manoj Kumar**[*]                                                       *manoj@iitmz.ac.in*
*School of Engineering and Science*
*Indian Institute of Technology Madras, Zanzibar Campus*

**Reviewed on OpenReview: https://openreview.net/forum?id=CxW6TF1rOF**

## Abstract

Neural ordinary differential equations (neural ODEs) represent a widely-used class of deep learning models characterized by continuous depth. Understanding the generalization error bound is important to evaluate how well a model is expected to perform on new, unseen data. Earlier works in this direction involved considering the linear case on the dynamics function (a function that models the evolution of state variables) of Neural ODE Marion (2023). Other related work is on bound for Neural Controlled ODE Bleistein & Guilloux (2023) that depends on the sampling gap. We consider a class of neural ordinary differential equations (ODEs) with a general nonlinear function for time-dependent and time-independent cases which is Lipschitz with respect to state variables. We observed that the solution of the neural ODEs would be of bounded variations if we assume that the dynamics function of Neural ODEs is Lipschitz continuous with respect to the hidden state. We derive a generalization bound for the time-dependent and time-independent Neural ODEs. We showed the effect of overparameterization and domain bound in the generalization error bound. This is the first time, the generalization bound for the Neural ODE with a general non-linear function has been found.

## 1 INTRODUCTION

Neural Ordinary Differential Equations (Chen et al. (2018)) are a class of deep learning models where the transformation between layers is treated as a continuous process defined by an ordinary differential equation (ODE). This idea generalizes the concept of residual networks (ResNets), where the evolution of the hidden state $z(t)$ over time is modeled by a differential equation

$$\frac{dz(t)}{dt} = f(z(t), t, \theta(t)) \quad \text{with} \quad z(0) = x, \tag{1.1}$$

where $\theta(t)$ represents the parameters of the model.

Unlike discrete representations from the conventional methods, neural ordinary differential equations (Neural ODEs) directly learn continuous latent representations (or latent states) based on a vector field parameterized by a neural network. Kidger et al. (2020) introduced neural controlled differential equations (Neural CDEs), which are continuous-time analogs of ResNets that use controlled paths to represent irregular time series. Neural ODEs are also extended to neural stochastic differential equations (Neural SDEs) with a focus on

---

[*]Both the authors contributed equally for this work.
Website https://sites.google.com/view/hansat
Code is open-source at : https://github.com/Madhusudan-Verma/Gen-bound-Node

aspects such as gradient computation, variational inference for latent spaces, and uncertainty quantification. In neural stochastic ODEs (neural SDEs, Oh et al. (2024)), usually a diffusion term is incorporated but a careful design of drift and diffusion term is essential.

With neural ODEs, generally, it is difficult to handle irregular time-series data. Neural controlled differential equations (Kidger et al. (2020)) generalize neural ODEs by incorporating a control mechanism, allowing them to model the evolution of hidden states as controlled differential equations. Studying the statistical properties of neural ODEs is not a trivial task. Since standard measures of statistical complexity in neural networks, such as those discussed by Bartlett et al. (2019), typically increase with depth, it is unclear why models with effectively infinite depth, like neural ODEs, would demonstrate strong generalization capabilities.

Marion (2023) studied the statistical properties of a class of time-dependent neural ODEs described by the following equation:

$$\frac{d\mathbf{H}_t}{dt} = \mathbf{W}_t \sigma(\mathbf{H}_t),$$

where $\mathbf{W}_t \in \mathbb{R}^{d \times d}$ is a weight matrix that depends on the time index $t$, and $\sigma : \mathbb{R} \to \mathbb{R}$ is an activation function applied component-wise. The model considered by Marion (2023) does not include the case where there are weights inside the non-linearity since they assume the dynamics at time $t$ to be linear with respect to the parameters.

**Contribution**. For a general class of well-posed neural ODEs, where the neural network involves the non-linear weights within the function, there is no result related to the generalization bound. In this work, we consider a Neural ODE model parameterized by $\theta(t)$ of the following form:

$$\frac{dz(t)}{dt} = f(z(t), t, \theta(t)) \quad \text{with} \quad z(0) = x, \tag{1.2}$$

where $f : \mathbb{R}^d \times \mathbb{R} \times \mathbb{R}^{n_\theta} \to \mathbb{R}^d$ , $z : [0, L] \to \mathbb{R}^d$ and $\theta : [0, L] \to \mathbb{R}^{n_\theta}$. Time dependent $\theta$ in neural ODEs is introduced by Massaroli et al. (2020b). We provide a generalization bound for the large class of parameterized ODEs instead of a linear class, the bound we provided here will hold for a linear class as well and is stricter than the earlier bounds for the linear class of functions. To the best of our knowledge, this is the first available bound for neural ODEs for this class of functions.

**Organization.** Section 1 is devoted to the introduction, and in Section 2, we discuss the related works. In section 3, we formulated the learning problem. In this section we also discuss some of the preliminaries and definitions that are crucial for understanding the problem setup. Section 4 is devoted to derive results related to generalization bounds. We performed numerical experiments in section 5. In the end, some concluding remarks are given in section 6.

## 2 RELATED WORKS

**Hybridizing deep learning and differential equations.** The fusion of deep learning with differential equations has recently garnered renewed interest, although the concept has been explored since the 1990s Rico-Martinez et al. (1992; 1994). A notable advancement was presented by Chen et al. (2018), where they introduced a model that learns a representation $\mathbf{u} \in \mathbb{R}^n$ by setting the initial condition $z(0) = \phi_{\theta(t)}(\mathbf{u})$ for the following ordinary differential equation (ODE):

$$\frac{dz(t)}{dt} = f(z(t), t, \theta(t))$$

where both $f$ and $\phi_{\theta(t)}$ are neural networks. The solution at the final time $t_1$, denoted $\mathbf{z}(t_1)$, is then utilized as input to a conventional machine learning model. This approach seamlessly integrates neural networks and ODEs, offering a robust framework for learning complex dynamical systems. Since then, several works have built on this idea, including theoretical advancements and practical applications as seen in Dupont et al. (2019); Chen et al. (2019); Finlay et al. (2020). For a more comprehensive overview, readers may refer to the reviews by Massaroli et al. (2020a) and Kidger (2022), which delve into the intersection of differential equations and deep learning.

**Generalization of Neural Controlled Differential Equations.** Bleistein & Guilloux (2023) used a Lipschitz-based argument to obtain a sampling-dependant generalization bound for neural controlled differential equations (NCDEs). The NCDE considered was of the following form:

$$dz_t = G_\psi(z_t)d\tilde{x}_t,$$

where $z_t \in \mathbb{R}^p$, and $G_\psi : \mathbb{R}^p \to \mathbb{R}^{p \times d}$ is neural network parametrized by $\psi$, also $\tilde{x}_t \in \mathbb{R}^p$ is continuous path. In this work, it is assumed that $(x_t)$ is Lipschitz which implies that $x = (x_t)_{t \in [0,1]}$ is of bounded variation. They also analyzed how approximation and generalization are affected by irregular sampling.

**Generalization bounds for neural networks.** Bartlett et al. (2017a) derived a margin-based multiclass generalization bound for neural networks that scales with margin-normalized spectral complexity, involving the Lipschitz constant (the product of the spectral norms of the weight matrices) and a correction factor. Long & Sedghi (2019) established generalization error bounds for convolutional networks based on training loss, parameter count, the Lipschitz constant of the loss, and the distance between current and initial weights, independent of input size and hidden layer dimensions. Experiments on CIFAR-10 show these bounds align with observed generalization gaps under varying hyperparameters in deep convolutional networks. Wang & Ma (2022) derive generalization error bounds for deep neural networks trained via SGD by combining control of parameter norms with Rademacher complexity estimates. These bounds, which apply to various architectures like MLPs and CNNs, depend on the training loss and do not require L-smoothness, making them more broadly applicable than stability-based bounds. Understanding generalization in neural networks has been approached through various theoretical frameworks, including VC-dimension (Vapnik & Chervonenkis, 1971) Vapnik & Chervonenkis (1971), Rademacher complexity (Bartlett & Mendelson, 2002) Bartlett & Mendelson (2002), PAC-Bayes theory (McAllester, 1999) McAllester (1999), compression-based bounds (Arora et al., 2018) Arora et al. (2018), stability analysis (Bousquet & Elisseeff, 2002; Hardt et al., 2016) Bousquet & Elisseeff (2002); Hardt et al. (2016), and information-theoretic approaches (Xu & Raginsky, 2017; Tishby & Zaslavsky, 2015) Xu & Raginsky (2017); Tishby & Zaslavsky (2015). While VC-dimension and covering number bounds are often loose for deep networks, norm-based and margin-based bounds (Bartlett et al., 2017; Neyshabur et al., 2015) Bartlett et al. (2017c); Neyshabur et al. (2015) offer sharper estimates. PAC-Bayes bounds have been effectively applied to deep models by optimizing the bound directly (Dziugaite & Roy, 2017) Dziugaite & Roy (2017).

## 3 BACKGROUND AND LEARNING PROBLEM

Let $h_\theta(x)$ be the solution of Neural ODE (1.1) at the final time $t = L$ ($L$ can also be taken as 1 for simplicity) with initial condition $x$ and let $y_i$ be the true label of the differential equation at the final time step. Also, assume that $\mathcal{H}_\theta$ be the set of all predictors $h_\theta$ with different initial conditions. We consider an i.i.d. sample $\{(y_i, x_i)\}_{i=1}^n \sim (y, x)$. For a given predictor $h_\theta$, the empirical risk over the training data is :

$$R^n(h_\theta) = \frac{1}{n} \sum_{i=1}^n \ell(y_i, h_\theta(x_i)).$$

The expected risk or generalization error over the data distribution is:

$$R(h_\theta) = \mathbb{E}_{(x,y) \sim \mathcal{P}}[\ell(y, h_\theta(x))] = \mathbb{E}[\ell(y, h_\theta)],$$

where $\mathcal{P}$ is the unknown, true probability distribution of the data. $R(h_\theta)$ cannot be optimized, since we do not have access to the continuous data. Let $\hat{\theta} \in \arg\min_\theta R^n(h_\theta))$ be an optimal parameter and $\hat{h}$ be the optimal predictor obtained by empirical risk minimization. In order to obtain generalization bounds, the following assumptions on the loss and the outcome are necessary Mohri (2018).

For input $z(t) \in \mathbb{R}^d$, with weights $A_i(t) \in \mathbb{R}^{m \times d}$ and biases $b_i(t) \in \mathbb{R}^m$ $for$ $i = 1, 2 \dots N$, we assume that the feed forward neural network with $N$ layers and $m$ number of hidden units takes the following form.

$$f(z(t), t, \theta(t)) := \sigma\left(A_N(t)\sigma\left(A_{N-1}(t)\sigma\left(\dots\sigma\left(A_1(t)z + b_1\right)\right) + b_{N-1}\right) + b_N\right). \tag{3.3}$$

$\sigma$ is the activation function, $\theta(t) = [A_1, A_2, A_3....A_N, b_1, b_2, b_3....b_N]$ where $z \in \mathbb{R}^d$. We have the following assumptions:

**Assumption 1.** $f(z(t), t, \theta(t))$ is assumed to be Lipschitz continuous with respect to $z(t)$, that means

$$|f(z_1, t, \theta(t)) - f(z_2, t, \theta(t))| \leq L_{f_{\theta(t)}}|z_1 - z_2|.$$

**Assumption 2.** Weights $A_i(t)$ and biases $b_i(t)$ of feed forward neural network defined in (3.3) are Lipschitz continuous.

**Assumption 3.** The outcome $y \in \mathbb{R}^d$ is bounded almost surely.

**Assumption 4.** The loss $\ell : \mathbb{R}^d \times \mathbb{R}^d \to \mathbb{R}_+$ is Lipschitz continuous with respect to its second variable, that is, there exists $L_\ell$ such that for all $u, u' \in Y$ and $y \in Y$

$$|l(y, u) - l(y, u')| \leq L_l|u - u'|.$$

Assumption 4 is satisfied for most of the classical loss functions, such as the mean squared error, as long as the outcome and the predictions are bounded. Table 1 gives the details about the notations used in this work and Table 2 gives the details about the notations used in Marion (2023) and Bleistein & Guilloux (2023).

| Symbol | Meaning |
|---|---|
| $z(t) \in \mathbb{R}^d$ | Hidden state / solution of the Neural ODE at time $t$ |
| $x$ | Input / initial condition, i.e., $z(0) = x$ |
| $y \in \mathbb{R}^d$ | True outcome (bounded, Assumption 3) |
| $\theta(t) \in \mathbb{R}^{n_\theta}$ | Time-dependent parameters of the Neural ODE |
| $f(z(t), t, \theta(t))$ | Dynamics function (vector field), assumed Lipschitz in $z$ |
| $L_{f_{\theta(t)}}$ | Lipschitz constant for $f(z(t), t, \theta(t))$ w.r.t. $z$ |
| $L$ | Final time horizon (ODE solved on $[0, L]$) |
| $d$ | Dimension of solution space |
| $h_\theta(x)$ | Predictor: solution of the Neural ODE at final time $t = L$ |
| $H_\theta$ | Hypothesis class of predictors |
| $R(h_\theta)$ | Expected risk (generalization error) |
| $R_n(h_\theta)$ | Empirical risk on $n$ samples |
| $\ell(y, \hat{y})$ | Loss function, assumed Lipschitz in prediction argument |
| $M_\ell$ | Upper bound on the loss function |
| $L_\ell$ | Lipschitz constant of the loss function |
| $\sigma$ | Activation function (e.g., ReLU, Tanh), $L_\sigma$-Lipschitz |
| $A_i(t), b_i(t)$ | Weight matrices and bias vectors of the feedforward network parameterizing $f$ |
| $L_A, L_b$ | Lipschitz constants of weights and biases (time-dependent case) |
| $V$ | Upper bound on the solution norm of Neural ODE |
| $\widehat{\mathcal{R}}_n(\mathcal{H})$ | Empirical Rademacher complexity of hypothesis class $\mathcal{H}$ |
| $N(\tau, \mathcal{H}, \rho)$ | Covering number of hypothesis class $\mathcal{H}$ under metric $\rho$ |
| $\mu$ | Lipschitz constant of the loss in its second argument |
| $\delta$ | Confidence parameter in generalization bounds |
| $\mathbb{P}$ | Data distribution |
| $\eta_i$ | Rademacher random variables ($\pm 1$ with equal probability) |

Table 1: Table of Notations

## 3.1 Preliminaries

In this subsection, we present essential preliminary results and definitions necessary for constructing the theoretical framework to derive the desired generalization bound. We begin with the definition of functions of bounded variation, as the solutions of the ordinary differential equations (ODEs) under consideration

| Symbol | Meaning |
|--------|---------|
| $m$ | Number of basis functions (Marion, 2023) |
| $R_\Theta$ | $\ell_{1,\infty}$ bound on parameter function $\theta$ (Marion, 2023) |
| $K_\Theta$ | Lipschitz constant of parameter function $\theta$ (Marion, 2023) |
| $F_\Theta$ | Class of parameterized ODEs (Marion, 2023) |
| $q$ | Depth of the neural network (number of layers) in $G_\psi$ (Bleistein & Guilloux, 2023) |
| $p$ | Width / dimension of hidden layers in $G_\psi$ (Bleistein & Guilloux, 2023) |
| $G_\psi(z)$ | Dynamics function parameterized by $\psi$ (Bleistein & Guilloux, 2023) |
| $A_h, b_h$ | Weight matrices and bias vectors at layer $h$ (Bleistein & Guilloux, 2023) |
| $U, v, \Phi$ | Additional network parameters in NCDE bound (Bleistein & Guilloux, 2023) |
| $K_1^D, K_2^D$ | Discretization and depth-dependent constants (Bleistein & Guilloux, 2023) |
| $M_\Theta$ | Upper bound on $\|G_\psi(0)\|_{op}$ (Bleistein & Guilloux, 2023) |

Table 2: Table of Notations (Prior Works: Marion (2023); Bleistein & Guilloux (2023))

are observed to exhibit this property. Next, we review several formulations of Gronwall's lemma, a critical tool for establishing bounds on ODE solutions. We then introduce the concept of covering numbers, which plays a pivotal role in bounding the Rademacher complexity. Finally, we summarize key results related to Rademacher complexity, enabling the subsequent derivation of the generalization bound.

**Definition 3.1** (Dutta & Nguyen (2018)). *The function $u \in L^1(\Omega, \mathbb{R})$ is a function of bounded variation on $\Omega$ (denoted by $BV(\Omega, \mathbb{R})$) if the distributional derivative of $u$ is representable by a finite Radon measure in $\Omega$, i.e., if*

$$\int_\Omega u \cdot \frac{\partial \varphi}{\partial x_i}\, dx = -\int_\Omega \varphi\, dD_i u \quad \text{for all } \varphi \in C_c^1(\Omega, \mathbb{R}),\ i \in \{1, 2, \ldots, n\},$$

*for some Radon measure $Du = (D_1 u, D_2 u, \ldots, D_n u)$. We denote by $|Du|$ the total variation of the vector measure $Du$, i.e.,*

$$|Du|(\Omega) \quad = \sup\left\{ \int_\Omega u(x)\, div(\phi)\, dx \,\middle|\, \phi \in C_c^1(\Omega, \mathbb{R}^n),\ \|\phi\|_{L^\infty(\Omega)} \le 1 \right\}.$$

**Lemma 3.2.** *(Particular case of Gronwall's Inequality) Let $I$ denote an interval of the real line of the form $[a, \infty)$ or $[a, b]$ or $[a, b)$ with $a < b$. Let $\alpha$, $\beta$, and $u$ be real-valued functions defined on $I$. Assume that $\beta$ and $u$ are continuous and that the negative part of $\alpha$ is integrable on every closed and bounded subinterval of $I$.*

*Assume that $\beta$ is non-negative, function $\alpha$ is non-decreasing. If $u$ satisfies the integral inequality*

$$u(t) \le \alpha(t) + \int_a^t \beta(s) u(s)\, ds, \qquad \forall t \in I,$$

*then*

$$u(t) \le \alpha(t) \exp\left( \int_a^t \beta(s)\, ds \right), \qquad t \in I.$$

**Lemma 3.3.** *(Gronwall's Lemma for sequences). Let $(y_k)_{k \ge 0}$, $(b_k)_{k \ge 0}$, and $(f_k)_{k \ge 0}$ be positive sequences of real numbers such that*

$$y_n \le f_n + \sum_{l=0}^{n-1} b_l y_l$$

*for all $n \ge 0$. Then*

$$y_n \le f_n + \sum_{l=0}^{n-1} f_l b_l \prod_{j=l+1}^{n-1} (1 + b_j)$$

*for all $n \ge 0$.*

Proof can be found in Holte (2009) and Clark (1987). We need a variant of Gronwall's Lemma for sequences.

**Lemma 3.4.** *Let $(u_k)_{k \geq 0}$ be a sequence such that for all $k \geq 1$,*

$$u_k \leq a_k u_{k-1} + b_k$$

*for $(a_k)_{k \geq 1}$ and $(b_k)_{k \geq 1}$ two positive sequences. Then for all $k \geq 1$,*

$$u_k \leq \left( \prod_{j=1}^{k} a_j \right) u_0 + \sum_{j=1}^{k} b_j \left( \prod_{i=j+1}^{k} a_i \right).$$

**Definition 3.5** (Bartlett et al. (2017b)). *Let $(M, \rho)$ be a metric space. A subset $\hat{T} \subseteq M$ is called an $\tau$-cover of $T \subseteq M$ if for every $m \in T$, there exists an $m' \in \hat{T}$ such that $\rho(m, m') \leq \tau$. $\hat{T}$ is called a proper cover if $\hat{T} \subset T$. The $\tau$ covering number of $T$ is the cardinality of the smallest $\tau$-cover of $T$, that is*

$$N(\tau, T, \rho) = \min\{|\hat{T}| : \hat{T} \text{ is an } \tau \text{ cover of } T\}.$$

**Lemma 3.6** (Gautschi (1959)). *For $x > 0$ and $0 < \lambda < 1$, the inequality holds*

$$x^{1-\lambda} \leq \frac{\Gamma(x+1)}{\Gamma(x+\lambda)} \leq (x+1)^{1-\lambda}.$$

**Rademacher Complexity :** Rademacher complexity is a concept from statistical learning theory that measures the richness of a class of functions in terms of how well they can fit random noise. It is commonly used to derive bounds on the generalization error of learning algorithms.

**Definition 3.7** (Mohri (2018)). *Given a class of functions $\mathcal{H}$ mapping from an input space $\mathcal{X}$ to $\mathbb{R}$ and a sample $S = \{x_1, x_2, \ldots, x_n\}$ drawn from a distribution $\mathcal{D}$, the empirical Rademacher complexity of $\mathcal{H}$ with respect to the sample $S$ is defined as:*

$$\hat{\mathcal{R}}_n(\mathcal{H}) = \mathbb{E}_\eta \left[ \sup_{h \in \mathcal{H}} \frac{1}{n} \sum_{i=1}^{n} \eta_i h(x_i) \right],$$

*where $\eta_i$ are independent Rademacher variables, taking values $+1$ or $-1$ with equal probability, and the expectation $\mathbb{E}_\eta$ is taken over the distribution of the Rademacher variables.*

**Lemma 3.8** (Srebro & Sridharan). *Let $(F_{x_1,\ldots,x_n}, L_2(P_n))$ denote the data-dependent $L_2$ metric space, given by the metric*

$$\rho(f, f') = \frac{1}{n} \sum_{i=1}^{n} (f(x_i) - f'(x_i))^2$$

*where $x_1, \ldots, x_n$ are samples from space $S$, and $F_{x_1,\ldots,x_n}$ denotes the restriction of the function class $F$ to that sample.*

*For any function class $\mathcal{F}$ containing functions $f : S \to \mathbb{R}$ and a sample $S = \{x_1, x_2, \ldots, x_n\}$ drawn from a distribution $\mathcal{D}$ we have*

$$\hat{\mathcal{R}}_n(\mathcal{F}) \quad \leq \quad \inf_{\epsilon \geq 0} \left\{ 4\epsilon + 12 \int_\epsilon^{\sup_{f \in \mathcal{F}} \sqrt{\mathbb{E}[\hat{f}^2]}} \sqrt{\frac{\log N(\tau, \mathcal{F}, L_2(P_n))}{n}} \, d\tau \right\}$$

*where $N(\tau, \mathcal{F}, L_2(P_n))$ denotes the covering number of $\mathcal{F}$.*

**Lemma 3.9** (Mohri (2018)). *Rademacher complexity regression bounds : Let*

$$\mathcal{H} = \{h : \mathcal{X} \to \mathcal{Y}\}$$

*be a set of functions from an input space to an output space that a learning algorithm can choose from during training. Let $l : \mathcal{Y} \times \mathcal{Y} \to \mathbb{R}$ be a non-negative loss function, upper bounded by $M > 0$ ($\ell(y, y') \le M$ for all $y, y' \in \mathcal{Y}$), and such that for any fixed $y' \in \mathcal{Y}$, the function $y \mapsto \ell(y, y')$ is $\mu$-Lipschitz for some $\mu > 0$ ($|l(y_1, y') - l(y_2, y')| \le \mu|y_1 - y_2| \ \ \forall y_1, y_2 \in \mathcal{Y}$). Let $\delta \in (0, 1)$, then with the probability at least $1 - \delta$.*

$$\mathbb{E}_{(x,y) \sim \mathcal{D}}\left[\ell(h(x), y)\right] \quad \le \quad \frac{1}{n}\sum_{i=1}^{n}\ell(h(x_i), y_i) + 2\mu\hat{\mathcal{R}}_n(\mathcal{H}) + 3M\sqrt{\frac{\log\frac{2}{\delta}}{2n}}.$$

# 4 MAIN RESULTS

We state and prove important lemmas before proceeding to the proof of the main Theorem 4.9. We assume that $f(z(t), t, \theta(t))$ is Lipschitz continuous with respect to $z$. So, by the Mean Value Theorem, the solution to equation (1.1) will be of bounded variation. More details can be found in appendix A.1.

**Lemma 4.1.** *Consider the feed forward neural network $f_N(z(t))$ defined in (3.3), assume that $\sigma$ is $L_\sigma$ Lipschitz, and $A_i$'s are bounded by $\mathcal{A}$ and biased terms are bounded by $\mathbf{B}$. Let $\|A_i(0)\| \le B_{A_0}$ , $\|b_i(0)\| \le B_{b_0}$, $t \in [0, L]$ and $L_A$ and $L_b$ are Lipschitz constant for weights and biases respectively.*

*Using equation A.5 we have, $\|A_i(t)\| \le \|A_i(0)\| + L_A L \le B_{A_0} + L_A L = \mathcal{A}$ and $\|b_i(t)\| \le \|b_i(0)\| + L_b L \le B_{b_0} + L_b L = \mathbf{B}$. Then,*

$$\|z(t)\| \le \left(\|z(0)\| + tL_\sigma\mathbf{B}\frac{(L_\sigma\mathcal{A}^N - 1)}{L_\sigma\mathcal{A}}\right)\exp\left(tL_{f_{\theta(t)}}\right).$$

**Corollary 4.2.** *In the case of time-independent Neural ODE Lipschitz constants for weights and biases will be 0, hence $B_{A_0} = \mathcal{A}$ and $B_{b_0} = \mathbf{B}$ and*

$$\|z(t)\| \le \left(\|z(0)\| + LL_\sigma B_{b_0}\frac{(L_\sigma B_{A_0}^N - 1)}{L_\sigma B_{A_0}}\right)\exp\left(LL_{f_\theta}\right). \tag{4.4}$$

*This bound on the solution will be useful to obtain the explicit form for covering number bound. This bound involves the Lipschitz constant, bound on biased terms and weights.*

Let $Dz$ be the distributional derivative of solution function $z$ and

$$\mathcal{I} = \{z \in L^1([0, L]) \mid z \text{ is non decreasing}\}$$

$$\mathcal{B} = \{z \in L^1([0, L]) \mid |Dz|((0, L)) \le M\}.$$

Then we have the following lemma:

**Lemma 4.3.** *Let*

$$V = \left(\|z(0)\| + LL_\sigma\mathbf{B}\frac{(L_\sigma\mathcal{A}^N - 1)}{L_\sigma\mathcal{A}}\right)\exp\left(LL_{f_{\theta(t)}}\right)$$

*and $0 < \tau \le \frac{LV}{\tau}$, then*

$$N(\tau, \mathcal{I}, L_2(P_n)) \le \frac{2^{4\frac{LV}{\tau}}}{18}.$$

**Remark 4.4.** *Observe that the covering number bound increases exponentially with domain size and the bound of solution. We obtain a stricter bound on covering number for the class of bounded variation functions in $L^1$ space , earlier bound can be found in Lemma 2.3, Dutta & Nguyen (2018).*

**Corollary 4.5.** *Let $0 < \tau \le \frac{LV}{\tau}$, then*

$$N(\tau, \mathcal{B}, L_2(P_n)) \le \frac{2^{16\frac{LV}{\tau}}}{324}.$$

*Proof.* From Dutta & Nguyen (2018), we know that

$$N(\tau, \mathcal{B}, L_2(P_n)) \le N^2\left(\frac{\tau}{2}, \mathcal{I}, L_2(P_n)\right),$$

which proves the required result. □

**Remark 4.6.** *In the above lemma, the bound is dependent on covering number of non-decreasing functions with the radius of balls getting half. But for the class of bounded variation functions, we do not assume that the functions are non-decreasing.*

**Lemma 4.7.** *Let $\mathcal{B}'$ be the class of $\mathbb{R}^d$ valued functions with domain $[0, L]$ that are of bounded variation, then*

$$\hat{\mathcal{R}}_n(\mathcal{B}') \le 96\frac{\sqrt{bLVd^{\frac{3}{2}}\log 2}}{\sqrt{n}} - 576\frac{LVd^{\frac{3}{2}}\log 2}{n}.$$

**Remark 4.8.** *Lemma 4.7 ensures that the bound on Rademacher complexity increases with the dimension of range space for bounded variation functions. Also, due to the constant $V$, we also get the dependence on weight parameters and Lipschitz constant of activation functions.*

**Theorem 4.9.** *(Generalization bound for Neural ODEs) Let $V$ be the upper bound of the solution of neural ODE, $d$ be the dimension of the solution, $\hat{h}$ be the optimal predictor and $h^*$ be the true solution and $L$ be the upper bound for time and $M > 0$ be an upper bound of non-negative loss function $l : [0, V] \times [0, V] \to \mathbb{R}$, i.e., $l(\hat{h}, h^*) \le M$ for all $\hat{h}, h^* \in [0, V]$ (assumption 3). Also, assume that for any fixed $\hat{h} \in [0, V]$, the mapping $y \mapsto l(\hat{h}, h^*)$ is $\mu$-Lipschitz for some $\mu > 0$ (assumption 4). Then generalization error is bounded with probability at least $1 - \delta$ by:*

$$R(\hat{h}) \quad \le \quad R^n(\hat{h}) + 2\mu\left(96\frac{\sqrt{bLVd^{\frac{3}{2}}\log 2}}{\sqrt{n}} - 576\frac{LVd^{\frac{3}{2}}\log 2}{n}\right) + 3M\sqrt{\frac{\log\frac{2}{\delta}}{2n}}.$$

**Outline of the proof.** We observed that the solution of the Neural ODEs described by equation 1.1 will be of bounded variations. We found stricter bound for covering number of this class of functions. We observed that the covering number is related to the number of positive integer solutions of an equation which is equal to central binomial coefficients. The central binomial coefficient obeys a recurrence relation which has a closed form solution. We then used inequality which the ratio of gamma functions satisfies. In this way, we obtained a stricter bound for the covering number of bounded variation functions. We assumed the parameters to be Lipschitz continuous and obtained a bound on weights and biases. We then found Rademacher complexity bound using Dudley's entropy integral stated in Lemma 3.8. Finally, we used the result for the Rademacher complexity in Lemma 4.7 to regression bound stated in Lemma 3.9. Since Rademacher complexity is non negative, $b \ge \frac{36LV\log 2}{n}$.

**Remark 4.10.** *Our analysis first establishes a stricter bound for increasing functions in the $L^1$ class, and by leveraging the relation between covering numbers of bounded variation functions in $L^1$ and those of increasing functions in $L^1$, we obtain a refined covering number bound for bounded variation functions. Unlike prior work that operates in parameter space, we conduct the analysis directly in solution space, which offers several advantages such as the resulting bounds depend on intrinsic trajectory properties (e.g., Lipschitz continuity) rather than parameter-norm constraints; moreover, solution-space bounds remain valid across different parameterizations and architectures that realize the same dynamics.*

## 4.1 Comparison with other bounds (Neural ODEs)

To highlight the novelty of our result, we compare it against the most closely related generalization bounds for Neural and parameterized ODEs. Table 3 summarizes the assumptions, asymptotic form of the bounds, and whether the dependence is solution-space or parameter-space. Unlike prior bounds that are inherently tied to parameter-space quantities (e.g., norms, Lipschitz constants, network depth), our result depends solely on *solution-space* properties. This shift broadens the applicability of the theory, since solution regularity can often be verified or controlled directly, even in models where parameter norms are hard to interpret or constrain.

| Work / Theorem | Assumptions | Generalization Bound |
|---|---|---|
| **Theorem 4.9 (This work)** | Bounded variation solution with bound $V$
Bounded loss with bound $M$
$\mu$-Lipschitz loss. | $R(\hat{h}) \leq R^n(\hat{h}) + 2\mu\left( \frac{96\sqrt{bLVd^{3/2}\log 2}}{\sqrt{n}} - \frac{576LVd^{3/2}\log 2}{n} \right)$ $+ 3M\sqrt{\frac{\log(2/\delta)}{2n}}$ |
| **Theorem 1 (Marion (2023))** | $\|\theta\|_{1,\infty} \leq R_\Theta$,
each $\theta_i$ $K_\Theta$-Lipschitz | $R(\hat{\theta}_n) \leq R^n(\hat{\theta}_n) + B\sqrt{\frac{(m+1)\log(R_\Theta mn)}{n}}$ $+ B\frac{m\sqrt{K_\Theta}}{n^{1/4}} + B\sqrt{\frac{\log(1/\delta)}{n}}$ |
| **Theorem 4.1 (Bleistein & Guilloux (2023))** | $q$-layer MLP,
Lipschitz activation,
bounded weights and biases. | $R_D(\hat{f}_D) - R^n(\hat{f}_D) \leq \frac{24MD_\Theta L_\ell}{\sqrt{n}}\sqrt{2pU_1^D + (q-1)p(p+1)U_2^D}$ $+ \frac{24MD_\Theta L_\ell}{\sqrt{n}}\sqrt{dp(2+p)U_3^D}$ $+ M_\ell\sqrt{\frac{\log(1/\delta)}{2n}}$ |

Table 3: Comparison of generalization bounds across different works.

| Constant | Definition / Meaning |
|---|---|
| | **Marion (2023)** |
| $R_\Theta$ | Radius of the parameter space $\Theta$ (parameter norm bound). |
| $K_\Theta$ | Uniform bound on parameter sensitivity of ODE solutions (controls stability w.r.t. $\theta$). |
| $B$ | Explicit constant: $6K_\ell K_f \exp(K_f R_\Theta)R_X + MR_\Theta \exp(K_f R_\Theta) + R_Y$. |
| | **Bleistein & Guilloux (2023)** |
| $U_1^D$ | $\log(\sqrt{n}C_q K_1^D)$, depends on first-layer weights. |
| $U_2^D$ | $\log(\sqrt{np}C_q K_2^D)$, depends on hidden-layer weights. |
| $U_3^D$ | $\log(\sqrt{ndp}C_q K_2^D)$, depends on bias norms. |
| $K_1^D$ | Constant depending on network weights: $\max\{B_\Phi MD_\Theta, B_v C_v\}$. |
| $K_2^D$ | Constant depending on architecture: $\max\{B_b C_b^D, B_A C_A^D, B_U C_U\}$. |
| $C_q$ | $(8q+12)$, depth-dependent scaling factor. |

Table 4: Definitions of constants appearing in generalization bounds in the works of Marion (2023), and Bleistein & Guilloux (2023).

The bound obtained in our work is tighter in terms of the sample size $n$ for the linear case, as it achieves a rate of $\mathcal{O}\big(n^{-1/2}\big)$ for the term involving the Lipschitz constant of the weights, whereas the corresponding rate in Marion (2023) is only $\mathcal{O}\big(n^{-1/4}\big)$. Moreover, while the bound in Marion (2023) is depth-independent but exhibits a worse dependence on the network width, our bound shows the opposite behavior: it depends on depth but not on width. In the bound given by Bleistein & Guilloux (2023), if we consider the case $x(t) = t$, which corresponds to a neural ODE, the dependence on $n$ matches ours; however, their bound depends on the discretization of time, whereas ours is independent of the discretization size. In addition, our bound is structurally simpler, as it involves fewer parameters.

## 5 NUMERICAL ILLUSTRATIONS

The experiment presented in Figure 1 investigates the impact of varying the number of hidden units in a Neural ODE model on generalization error. Neural ODEs, which model continuous transformations of data using ordinary differential equations, allow for varying model capacity by adjusting the number of hidden units in their ODE block. The input data, denoted as $\mathbf{X}$, consists of $n_{\text{samples}}$ random vectors, each with a dimensionality of input_dim, and is drawn from a standard normal distribution:

$$\mathbf{X} \sim \mathcal{N}(0, 1)_{n_{\text{samples}} \times 2}.$$

The target values, $\mathbf{y}$, are generated by applying a sinusoidal transformation to the sum of elements in each input vector, where

$$y_i = \sin\left(\sum_{j=1}^{2} X_{ij}\right) \quad \text{for all} \quad i \in \{1, 2, \ldots, n_{\text{samples}}\}.$$

This non-linear transformation introduces complexity into the data, while keeping the output values within a bounded range.

By adjusting the number of hidden units in the ODE block, we analyze how the model's capacity affects its generalization ability, defined as the difference in error on unseen test data after training. The results show that as the number of hidden units increases, the generalization error also increases. This observation empirically validates Theorem 4.9 , which suggests that the generalization gap is influenced by the norm bound of the network parameters. As the number of hidden units grows, the norm of the weight matrices also increases, amplifying the Lipschitz constant of the transformation and making the model more sensitive to small input perturbations. This heightened sensitivity reduces robustness and leads to a larger generalization gap. Thus, the experiment provides empirical confirmation of the theoretical prediction, demonstrating that increasing model capacity leads to a higher generalization error due to the growing norm bound for weights.

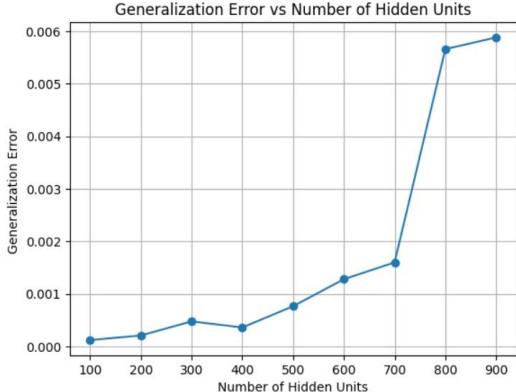

Figure 1: Generalization Error vs Number of Hidden Units in Neural ODE.

We conducted experiments on the MNIST and CIFAR-10 datasets to investigate the relationship between the Lipschitz constant of the dynamics function in a Neural ODE and its generalization performance. In

both cases, we trained a Neural ODE model in which the feature dynamics are governed by a two-layer fully connected neural network with `Tanh` activation. The dynamics function $f$ was defined as $f(z,t) = W_2 \cdot \tanh(W_1 z)$, and its Lipschitz constant was approximated by $\|W_2\|_2 \cdot \|W_1\|_2$, where $\|\cdot\|_2$ denotes the spectral norm. At each training epoch, we recorded the model's training and test accuracies, computed the generalization gap (test error minus train error), and measured the Lipschitz constant of the dynamics.

The plots in Figure 2 show that for both MNIST and CIFAR-10, the generalization gap increases with the Lipschitz constant of the dynamics function. This positive correlation is stronger for CIFAR-10, which is a more complex dataset than MNIST. The findings suggest that Neural ODEs with higher Lipschitz constants (i.e., more sensitive dynamics) tend to overfit the training data and thus generalize poorly. This behavior aligns with theoretical expectations from generalization bounds for Lipschitz-continuous functions, which suggest that the generalization gap scales proportionally with the Lipschitz constant:

$$\text{Gen Gap} \lesssim \|f\|_{\text{Lip}} \cdot \mathcal{O}\left(\frac{1}{\sqrt{n}}\right).$$

These results highlight the importance of controlling the Lipschitz continuity of the learned dynamics in Neural ODEs to ensure good generalization. Moreover, they provide empirical support for the theoretical notion that models with smoother transformations (lower Lipschitz constants) are less prone to overfitting.

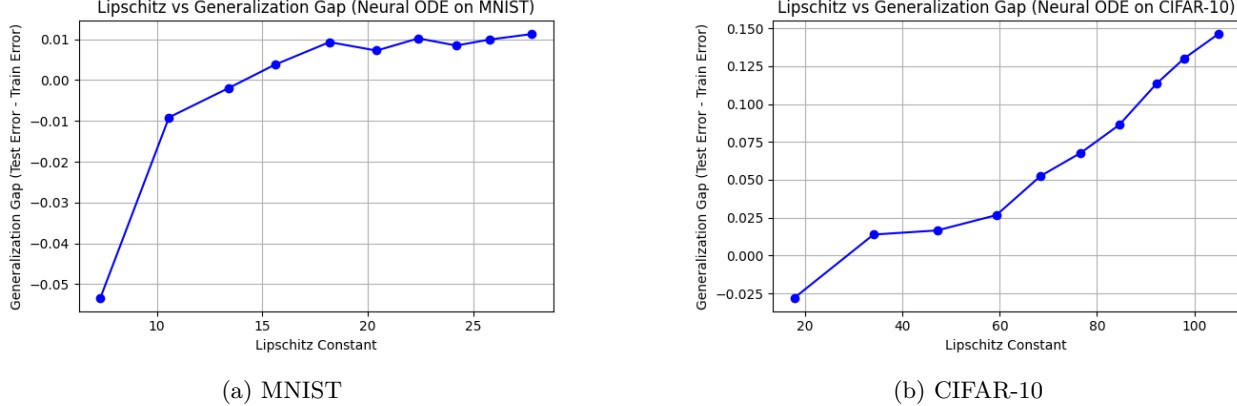

(a) MNIST  (b) CIFAR-10

Figure 2: Generalization gap vs Lipschitz constant for Neural ODEs on MNIST and CIFAR-10.

The experiment shown by figure 3 is to investigate how the generalization gap is related to the Lipschitz constant of weights $\sup_{0 \leq k \leq L-1} \|W_{k+1} - W_k\|$. The Neural ODE is defined with time-varying weights, where the forward pass involves applying a sinusoidal time dependency to the weights of the hidden layer. The model computes the Lipschitz constant by calculating the largest singular value of the weight matrices, which serves as a measure of how sensitive the model is to input changes. Lipschitz constant of weights is added as a penalty term in the loss function with different regularization parameters ($\lambda$ )values. The results are summarized in a box plot, showing the generalization gap versus $\lambda$, to visualize the impact of varying the penalization factor on the model's generalization performance. As we increase the value of the penalization factor the average generalization gap decreases which is possible only if Lipschitz constant of weights decreases which indicates models with less Lipschitz constant of weights have less generalization gap. So, the Lipschitz constant of weights should be in the numerator, which is confirmed by theoretical bound.

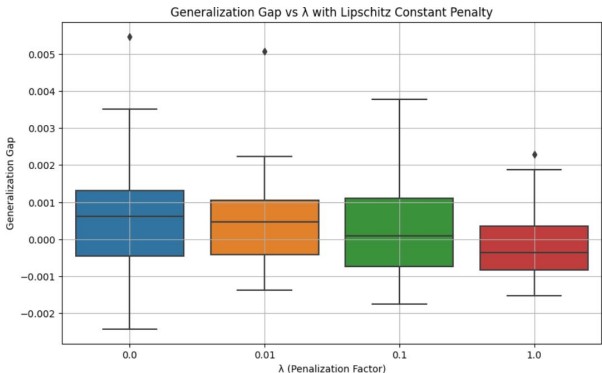

Figure 3: Plot of generalization gap against regularization parameter for time dependent Neural ODE. Lipschitz constant for weights which is in the bound of solution is added as a penalty term to the loss function. Four different $\lambda$ values (0, 0.01, 0.1, and 1) are tested over 20 trials. For each trial, the generalization gap is calculated as the difference between the validation loss and training loss after training for 50 epochs.

An important implication of our generalization bounds is the explicit dependence on the Lipschitz constant $L_{f_{\theta(t)}}$ of the Neural ODE's vector field. Since larger $L_{f_{\theta(t)}}$ leads to looser generalization guarantees, one can improve robustness by explicitly penalizing high Lipschitz constants during training. This connects theory to practice, reducing the Lipschitz constant not only promotes stability of the learned dynamics but also directly tightens the theoretical bound. A simple strategy is to add a regularization term Jacobian norm $\|\nabla_z f_\theta(z(t), t)\|$ as used in Josias & Brink (2022). Alternative proxies include spectral norm constraints on weight matrices or layer-wise gradient penalties, both of which upper-bound the Lipschitz constant.

## 6 CONCLUSION AND OUTLOOK

In this work, we present the first generalization bounds for both time-independent and time-dependent neural ODEs. Specifically, we derived generalization bounds for time-dependent neural ODEs of the form

$$dz(t) = f(z(t), t, \theta(t))\, dt,$$

and, leveraging this result, we also obtained the corresponding generalization bounds for time-independent neural ODEs. One of the key insights of our analysis is the dependence of the generalization gap on the Lipschitz constant of the weights in the case of time-dependent neural ODEs. This relationship highlights the crucial role of the model's smoothness and stability in its ability to generalize to unseen data. An important open direction is to determine the tightness of the error bounds derived in this work. In particular, it remains unclear whether the dependence on the key quantities such as sample size, time horizon and state-Lipschitz constant of dynamic function of Neural ODE can be improved without imposing additional structural assumptions. To the best of our knowledge, matching minimax lower bounds for the general Neural ODE class are not available in the literature. Future work could therefore focus on (i) establishing lower bounds for natural subclasses of Neural ODEs (e.g., vector fields with prescribed smoothness, bounded Jacobians, or contractive dynamics), and (ii) performing systematic scaling experiments, varying time horizon and state-Lipschitz constant of dynamic function of Neural ODE , and architectural choices, to assess whether the theoretical exponents and constants align with empirical generalization behavior.

We recognize that stochastic neural ODEs, which have been shown to emerge as the deep limit of a wide class of residual neural networks, provide a natural extension to our results. Exploring generalization bounds for Neural Stochastic Differential Equations (Neural SDEs) presents an exciting avenue for future work, as it would further deepen our understanding of the connection between stochastic dynamics and generalization in deep learning models. Another promising direction is the application of the mean-field approach, which could offer tighter and more accurate estimates for generalization bounds in both time-independent and

time-dependent settings. By extending our analysis to these more complex models, we hope to contribute to a more comprehensive theory of generalization for neural ODEs and their stochastic counterparts.

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

## Appendix

**Organization of the Appendix:** Section A in appendix provides the proofs for three lemmas which we used to prove the main theorem. We proved lemma 4.1, 4.3, and 4.7 in this section. Section B is devoted to details related to numerical experiments.

## A  Proofs

### A.1  Proof of bounded variation solutions:

Assume $f(z(t), t, \theta(t))$ is Lipschitz in $z$ with constant $L_{f_{\theta(t)}}$ as stated in assumption 1, i.e.,

$$|f(z_1, t, \theta(t)) - f(z_2, t, \theta(t))| \le L_{f_{\theta(t)}} |z_1 - z_2|.$$

Assume $f(z(t), t, \theta(t))$ is continuous in $t$ on the compact rectangle $[0, 1] \times [a, b]$. By continuity on a compact set, $f(z(t), t, \theta(t))$ is bounded:

$$|f(z, t, \theta(t))| \le M \quad \text{for some } M > 0.$$

The solution $z(t)$ satisfies

$$\frac{dz}{dt} = f(z(t), t, \theta(t)) \quad \text{with } z(0) = x.$$

By the Picard-Lindelöf theorem, $z(t)$ exists uniquely and is absolutely continuous.

Since

$$\left| \frac{dz}{dt} \right| = |f(z(t), t, \theta(t))| \le M,$$

the solution $z(t)$ is Lipschitz continuous with constant $M$:

$$|z(t_i) - z(t_{i-1})| \le M|t_i - t_{i-1}| \quad \forall t_i, t_{i-1} \in [0, 1].$$

For any partition $0 = t_0 < t_1 < \cdots < t_n = 1$, apply the Mean Value Theorem (MVT) to $z(t)$:

$$z(t_i) - z(t_{i-1}) = \frac{dz}{dt}(c_i) \cdot (t_i - t_{i-1}) \quad \text{for some } c_i \in (t_{i-1}, t_i).$$

Summing the absolute differences gives the total variation:

$$\sum_{i=1}^{n} |z(t_i) - z(t_{i-1})| = \sum_{i=1}^{n} |f(z(c_i), c_i, \theta(c_i))| \cdot (t_i - t_{i-1}).$$

Since $|f(z(c_i), c_i, \theta(c_i))| \le M$, the total variation is bounded by:

$$V_0^1(z) \le M \sum_{i=1}^{n} (t_i - t_{i-1}) = M \cdot (1 - 0) = M < \infty.$$

The solution $z(t)$ has finite total variation $V_0^1(z) \le M$, so it is of bounded variation. The Lipschitz continuity of $f(z(t), t, \theta(t))$ in $z$, combined with its boundedness due to continuity in $t$, ensures the result.

### A.2  Proof of lemma 4.1

*Proof.* Let $A(t)$ be a time-dependent matrix. We assume that $A(t)$ is *Lipschitz continuous* as stated in assumption 2, meaning there exists a constant $L_A$ such that for all $t_1, t_2 \in [t_0, t_f]$:

$$\|A(t_1) - A(t_2)\| \le L_A |t_1 - t_2|$$

where $L_A$ is the *Lipschitz constant* and $\| \cdot \|$ is a suitable matrix norm (e.g., Frobenius norm or operator norm).

To express $A(t)$ as a function of its initial value $A(t_0)$, we use the integral representation:

$$A(t) = A(t_0) + \int_{t_0}^{t} \frac{dA(s)}{ds} \, ds$$

where $\frac{dA(s)}{ds}$ is the time derivative of $A(s)$, and the integral captures the accumulation of changes over time.

Using the assumption that $A(t)$ is Lipschitz continuous, the time derivative $\frac{dA(s)}{ds}$ is bounded by the Lipschitz constant $L_A$. Therefore, for $s \in [t_0, t_f]$, we have:

$$\left\| \frac{dA(s)}{ds} \right\| \leq L_A$$

Substituting this bound into the integral representation of $A(t)$:

$$\|A(t) - A(t_0)\| \leq \int_{t_0}^{t} \left\| \frac{dA(s)}{ds} \right\| ds \leq \int_{t_0}^{t} L_A \, ds$$

This simplifies to:

$$\|A(t) - A(t_0)\| \leq L_A |t - t_0|$$

Thus, we have the bound:

$$\|A(t)\| \leq \|A(t_0)\| + L_A |t - t_0|$$

Finally, to remove the time dependency, we maximize the bound over the interval $[t_0, t_f]$:

$$\|A(t)\| \leq \|A(t_0)\| + L_A(t_f - t_0)$$

Thus, the matrix $A(t)$ is uniformly bounded by a time-independent constant $M_A$:

$$\|A(t)\| \leq M_A = \|A(t_0)\| + L_A(t_f - t_0)$$

Since $t \in [0, L]$,

$$M_A = \|A(0)\| + L_A L \tag{A.5}$$

For $f(z(t), t, \theta(t))$ defined in 3.3 let us first consider the case when d=1.
Let $Dz$ be the distributional derivative of solution function $z$ and

$$\mathcal{I} = \{z \in L^1([0, L]) \mid z \text{ is non decreasing}\}$$

$$\mathcal{B} = \{z \in L^1([0, L]) \mid |Dz|((0, L)) \leq M\}.$$

We know that finding solution to neural ODE (4.1) is equivalent to finding solution to the integral equation

$$z(t) = z(0) + \int_0^t f(z(t), t, \theta(t)) \, dt. \tag{A.6}$$

Taking norms, this yields:

$$\|z(t)\| \leq \|z(0)\| + \int_0^t \|f(z(t), t, \theta(t))\| \, dt. \tag{A.7}$$

Notice that since we assumed $f$ is Lipschitz with respect to $z$, we have that for all $z \in \mathbb{R}^d$:

$$\|f(z(t), t, \theta(t))\| \leq \|f(z(t), t, \theta(t)) - f(0, t, \theta(t))\| + \|f(0, t, \theta(t))\| \tag{A.8}$$

$$\leq \|f(z(t), t, \theta(t)) - f(0, t, \theta(t))\| + \|f(0, t, \theta(t))\| \tag{A.9}$$

$$\leq L_f \|z(t)\| + \|f(0, t, \theta(t))\| \tag{A.10}$$

where the last inequality follows from the fact that $f$ is Lipschitz. It follows that:

$$\|z(t)\| \le \|z(0)\| + \int_0^t \left( L_f \|z\| + \|f(0, t, \theta(t))\| \right) dt. \tag{A.11}$$

Using the fact that $\int_0^t dt = t$, one gets:

$$\|z(t)\| \le \|z(0)\| + t\|f(0, t, \theta(t))\| + L_f \int_0^t \|z(t)\| dt \tag{A.12}$$

Applying Gronwall's inequality stated in Lemma B.1.1 yields,

$$\|z(t)\| \le \left( \|z(0)\| + t\|f(0, t, \theta(t))\| \right) \exp(tL_f). \tag{A.13}$$

Let $\|A_i(0)\| \le B_{A_0}$ and $\|b_i(0)\| \le B_{b_0}$

Then using equation A.5 we get, $\|A_i(t)\| \le \|A_i(0)\| + L_A L \le B_{A_0} + L_A L = \mathcal{A}$ and $\|b_i(t)\| \le \|b_i(0)\| + L_b L \le B_{b_0} + L_b L = \mathbf{B}$

Since

$$\|f_N(0)\| = \|f_N(0) - \sigma(0)\| \le L_\sigma \|A_N(t) f_{N-1}(0)\| + L_\sigma \mathbf{B}, \tag{A.14}$$

$$\le L_\sigma \mathcal{A} \|f_{N-1}(0)\| + L_\sigma \mathbf{B}. \tag{A.15}$$

Using lemma 3.4,

$$\|f_N(0)\| \le (L_\sigma \mathcal{A})^{N-1} \|\sigma(b_1)\| + L_\sigma \mathbf{B} \sum_{j=0}^{N-2} (L_\sigma \mathcal{A})^j, \tag{A.16}$$

$$\le L_\sigma \mathbf{B} \sum_{j=0}^{N-1} (L_\sigma \mathcal{A})^j, \tag{A.17}$$

$$= L_\sigma \mathbf{B} \frac{(L_\sigma \mathcal{A})^N - 1}{L_\sigma \mathcal{A} - 1}. \tag{A.18}$$

This implies

$$\|z(t)\| \le \left( \|z(0)\| + t L_\sigma \mathbf{B} \frac{(L_\sigma \mathcal{A}^N - 1)}{L_\sigma \mathcal{A}} \right) \exp(tL_f). \tag{A.19}$$

$\square$

## A.3 Proof of lemma 4.3

*Proof.* For a fixed positive integer $n$, let us set the discretization size as $\Delta x = \frac{L}{n}$, $\Delta y = \frac{V}{n}$. To each $z \in \mathcal{I}$, we associate the pair of functions $(\psi^+[z], \psi^-[z])$ defined by

$$\overline{\psi^+}[z] = \sum_{k=0}^{N-1} \overline{\psi_k^+} \cdot \mathbf{I}[k \cdot \Delta x, (k+1) \cdot \Delta x], \tag{A.20}$$

where

$$
\begin{aligned}
\psi_k^- &= \left[\frac{z(k \cdot \Delta x + 0)}{\Delta y}\right], \\
\psi_k^+ &= \left[\frac{z((k+1) \cdot \Delta x - 0)}{\Delta y}\right] + 1.
\end{aligned}
$$

For $\mathcal{X}^{\overline{+}} \in \mathcal{I}$, define

$$U(\mathcal{X}^-, \mathcal{X}^+) = \{z \in \mathcal{I} \mid \mathcal{X}^- \le z \le \mathcal{X}^+\}.$$

Since $z \in U(\mathcal{X}^-[z], \mathcal{X}^+[z])$, the set

$$\mathcal{U} = \{U(\mathcal{X}^-[z], \mathcal{X}^+[z]) \mid f \in \mathcal{I}\}$$

is a covering of $\mathcal{I}$.

Since

$$\#\mathcal{U} \le \{0 \le a_0 \le a_1 \le \cdots \le a_{N-1} \le N \mid (a_k \in \mathbb{N})\}^2$$

and

$$
\begin{aligned}
&\#\{0 \le a_0 \le a_1 \le \cdots \le a_{N-1} \le N \mid (a_k \in \mathbb{N})\} \\
={}& \left\{(p_1, \ldots, p_{N+1}) \in \mathbb{N}^{N+1} \mid p_1 + \cdots + p_{N+1} = N\right\} \\
={}& \binom{2N}{N},
\end{aligned}
$$

the covering number for the class of functions in $\mathcal{I}$ is bounded by $\binom{2n}{n}^2$. Consider sums of powers of binomial coefficients: $a_n^r = \sum_{k=0}^n \binom{n}{k}^r$. For $r = 2$, the closed-form solution is given by

$$a_n^{(2)} = \binom{2n}{n}$$

i.e., the central binomial coefficients. $a_n^{(2)}$ obeys the recurrence relation

$$(n+1)a_{n+1}^{(2)} - (4n+2)a_n^{(2)} = 0.$$

After solving the recurrence relation we get,

$$\binom{2n}{n} = C_1 \frac{4^{n-1}}{\Gamma(n+1)} \left(\frac{3}{2}\right)_{2n-1}$$

$$((x)_n \text{ denotes Pochhammer symbol.})$$

$$= 2 \cdot \frac{2^{2(n-1)}}{\Gamma(n+1)} \left(\frac{3}{2}\right)_{2n-1}$$

(since $C_1 = 2$, which we can

obtain by setting $n = 0$ in previous equation.)

$$= \frac{2^{2(n-1)}}{\Gamma(n+1)} \frac{\Gamma(\frac{3}{2}+n-1)}{\Gamma(\frac{3}{2})}$$

$$= \frac{2^{2(n-1)}}{\Gamma(n+1)} \frac{\Gamma(n+\frac{1}{2})}{\sqrt{\frac{\pi}{2}}}$$

$$= \frac{2^{2(n-1)}}{\sqrt{\frac{\pi}{2}}} \frac{\Gamma(n+\frac{1}{2})}{\Gamma(n+1)}$$

$$= \frac{2^{2n}}{\sqrt{\pi}} \frac{\Gamma(n+\frac{1}{2})}{\Gamma(n+1)}$$

$$\leq \frac{2^{2n}}{\sqrt{\pi}} \frac{1}{\sqrt{n}} (\text{using Lemma 3.6})$$

$$= \frac{2^{2n}}{\sqrt{n\pi}}.$$

$$\implies \binom{2n}{n}^2 \leq \frac{2^{4n}}{n\pi}$$

$$\leq \frac{2^{4n}}{6\pi} (\text{if } n \geq 6)$$

$$\leq \frac{2^{4n}}{18}.$$

Let $n = \left\lceil \frac{LV}{\tau} \right\rceil + 1$, then

$$N(\tau, \mathcal{I}, L_2(P_n)) \leq \frac{2^{4\frac{LV}{\tau}}}{18}.$$

$\square$

### A.4  Proof of lemma 4.7

*Proof.* Since,

$$N(\tau, \mathcal{B}', L_2(P_n)) \leq \frac{2^{16\frac{LV}{\tau}}}{324}.$$

For $z \in \mathbb{R}^d$ ,

$$N(\tau, \mathcal{B}', L_2(P_n)) \leq \left(\frac{2^{16\frac{LV\sqrt{d}}{\tau}}}{324}\right)^d$$

Observe that,

$$\sqrt{\log N(\tau, \mathcal{B}', L_2(P_n))} \leq \frac{4\sqrt{LVd^{\frac{3}{2}}\log 2}}{\sqrt{\tau}} = g(\tau).$$

Therefore,

$$\int_a^b g(\tau)d\tau = 8\sqrt{LVd^{\frac{3}{2}}\log 2}\left[\sqrt{b}-\sqrt{a}\right] \tag{A.21}$$

We know that from Lemma (3.8) that empirical Rademacher Complexity $\hat{\mathcal{R}}_n(\mathcal{B}')$ has the following bound

$$\hat{\mathcal{R}}_n(\mathcal{B}') \le \inf_{\epsilon\ge 0}\left\{4\epsilon + 12\int_\epsilon^b \sqrt{\frac{\log N(\tau,\mathcal{B}',L_2(P_n))}{n}}d\tau\right\},$$

where $b = \sup_{f\in\mathcal{B}'}\sqrt{\mathbb{E}[f^2]}$. Using (A.21), we have

$$\hat{\mathcal{R}}_n(\mathcal{B}') \le \inf_{\epsilon\ge 0}\left\{4\epsilon + \frac{96\sqrt{LVd^{\frac{3}{2}}\log 2}}{\sqrt{n}}\left[\sqrt{b}-\sqrt{\epsilon}\right]\right\}.$$

This implies

$$\hat{\mathcal{R}}_n(\mathcal{B}') \le 96\frac{\sqrt{bLVd^{\frac{3}{2}}\log 2}}{\sqrt{n}} - 576\frac{LVd^{\frac{3}{2}}\log 2}{n}.$$

$\square$

# B  Experiment details

## B.1  For experiment illustrated by figure 1

The objective of this experiment is to analyze the effect of the number of hidden units on the generalization error of a Neural ODE model. The generalization error is defined as the model's performance on unseen test data, measured using the mean squared error (MSE). The study investigates the relationship between model complexity, as determined by the number of hidden units, and its ability to generalize.

The dataset is synthetically generated and consists of training and testing samples. The training set comprises 100 samples, while the test set includes 30 samples. Each input sample has two features, sampled from a standard normal distribution. The target values are computed using a non-linear function of the inputs with some added randomness. This introduces a non-linear relationship between inputs and targets, mimicking the challenges of real-world data.

The Neural ODE model used in this experiment consists of three main components. First, a linear input layer maps the input data into a higher-dimensional space determined by the number of hidden units. Second, the ODE function models the dynamics of the hidden state using a fully connected layer with ReLU activation, solving the ODE using the 'torchdiffeq.odeint' solver over the time interval $[0.0, 1.0]$. The final state of the ODE solver is passed through an output layer to produce the scalar prediction.

The independent variable in this study is the number of hidden units, which is varied across the following values: $[100, 200, 300, 400, 500, 600, 700, 800, 900]$. For each configuration, the model is trained for 100 epochs using the Adam optimizer with a learning rate of 0.01. The loss function used is the mean squared error (MSE), and the training process is conducted on a GPU if available. The dependent variable is the generalization error, which is evaluated as the mean squared error on the test dataset.

Reproducibility is ensured by setting random seeds for both `torch` and `numpy`. The model performance is evaluated by calculating the MSE on the training and test datasets after training. The generalization error is analyzed as a function of the number of hidden units, and a line plot is generated to visualize this relationship. The x-axis represents the number of hidden units, and the y-axis represents the corresponding generalization error.

The hypothesis of the experiment is that increasing the number of hidden units will initially reduce the generalization error as the model's capacity improves. However, beyond a certain point, overfitting may occur, leading to an increase in the generalization error. The experiment is designed to identify this trend and explore the optimal model complexity for the given task.

### B.1.1 Data Generation

The dataset used in this experiment is synthetically generated to test the Neural ODE model's capability to generalize to unseen data. The process creates input-output pairs based on random input vectors and a non-linear transformation for the target values. This ensures that the task is sufficiently challenging while allowing for reproducibility.

The steps for generating the data are as follows:

1. The input data, denoted as $X$, is a set of $n_{\text{samples}}$ random vectors, where each vector has a dimensionality of input_dim. The elements of $X$ are drawn from a standard normal distribution:

$$X \sim \mathcal{N}(0,1)^{n_{\text{samples}} \times 2}.$$

2. The target values, denoted as $y$, are generated by applying a sinusoidal transformation to the sum of the elements in each input vector:

$$y_i = \sin\left(\sum_{j=1}^{2} X_{ij}\right), \quad \forall i \in \{1, 2, \ldots, n_{\text{samples}}\}.$$

This non-linear transformation introduces complexity into the data while ensuring a bounded range for the output values.

3. The inputs $X$ and the corresponding targets $y$ are paired together to form the dataset:

$$\text{Dataset} = \{(X_i, y_i)\}_{i=1}^{n_{\text{samples}}}.$$

4. Two datasets are generated:
   - A training dataset with $n_{\text{samples}} = 100$.
   - A testing dataset with $n_{\text{samples}} = 30$.

   Both datasets are created independently using the same generation process to ensure the test data remains unseen during training.

5. The generated data is stored as PyTorch tensors, making it compatible with the Neural ODE model. This enables efficient data loading and processing during training and evaluation.

This synthetic data generation process provides a controlled setup for evaluating the generalization capabilities of the Neural ODE model. The use of a sinusoidal target function introduces a non-trivial learning problem while maintaining interpretability and ease of reproducibility.

### B.2 For experiment illustrated by figure 2

We conducted experiments on both the MNIST and CIFAR-10 datasets to investigate the relationship between the Lipschitz constant of the dynamics function in a Neural ODE and its generalization gap. The MNIST dataset consists of 60,000 training and 10,000 testing grayscale images of handwritten digits, each of size $28 \times 28$. The CIFAR-10 dataset includes 50,000 training and 10,000 testing color images of size $32 \times 32$ with three RGB channels.

For both datasets, we used a Neural ODE model in which the hidden state evolves according to a learned ordinary differential equation (ODE). The ODE's right-hand side function $f(z(t), t, \theta(t))$ is implemented as a two-layer fully connected neural network with `Tanh` activation. The solution at time $t = 1$ is passed through a linear classifier to produce class logits. The Lipschitz constant of $f$ was estimated by computing an upper bound given by the product of the spectral norms (operator $L_2$ norms) of the two weight matrices, i.e., $\|A_2\|_2 \cdot \|A_1\|_2$.

Training was performed for 10 epochs using the Adam optimizer with a learning rate of $1 \times 10^{-3}$ and cross-entropy loss. A batch size of 128 was used for both training and evaluation. At each epoch, we measured the training and test accuracies, computed the generalization gap (defined as test error minus training error), and recorded the Lipschitz constant. To ensure reproducibility, all sources of randomness were controlled by fixing random seeds. Finally, we plotted the generalization gap against the Lipschitz constant. In both datasets, we observed a consistent empirical trend showing that larger Lipschitz constants correspond to higher generalization gaps, suggesting that increased sensitivity in the learned dynamics function may degrade generalization performance.

### B.3   For experiment illustrated by figure 3

This experiment investigates the impact of Lipschitz regularization on the generalization gap in Neural Ordinary Differential Equation (ODE) models. A Neural ODE model is implemented where the parameters of the ODE depend on time. The primary goal is to examine how adding a penalty term proportional to the Lipschitz constant of the model's weights influences the generalization gap, which is defined as the difference between validation loss and training loss.

The Neural ODE model consists of an ODE function with two fully connected layers. The first layer maps 2-dimensional input data to a hidden representation of size 50 with ReLU activation. The second layer projects this representation back to a 2-dimensional output. To incorporate time-dependency, the hidden layer's output is modulated by a sine function of time, introducing a dynamic weight adjustment. The ODE is solved using the `odeint` function from the `torchdiffeq` library over a fixed time interval of $[0, 1]$.

To measure and regulate the Lipschitz constant of the model, the singular values of the weight matrices are computed. The Lipschitz constant is defined as the maximum singular value across all weight matrices. During training, the loss function combines the mean squared error (MSE) between model predictions and ground truth labels with a penalty term proportional to the Lipschitz constant. The overall loss is expressed as:

$$\text{Loss} = \text{MSE} + \lambda \cdot \sup_{0 \leq k \leq N-1} \|A_{k+1} - A_k\|$$

where $\lambda$ is the regularization strength, and $\sup_{0 \leq k \leq N-1} \|A_{k+1} - A_k\|$ is the Lipschitz constant of weights.

The datasets for training and validation are synthetically generated. Both datasets consist of 2-dimensional samples drawn from a standard normal distribution, $\mathcal{N}(0, 1)$. The training dataset contains 100 samples, while the validation dataset contains 20 samples. The corresponding labels are generated by scaling the input data by a factor of 2, resulting in a simple linear relationship. This ensures a clear evaluation of the model's generalization capabilities.

#### B.3.1   Data Generation

In this experiment, the input data and corresponding labels are synthetically generated to evaluate the generalization capability of a neural ODE model with a Lipschitz constant penalty. The input data consists of random 2-dimensional points, generated independently from a standard normal distribution. Specifically, for each input data point $x = (x_1, x_2)$, both features $x_1$ and $x_2$ are independently drawn from the standard normal distribution $\mathcal{N}(0, 1)$. This ensures that the dataset contains diverse points distributed across the 2-dimensional space. The dataset used for training consists of 100 such points, and the dataset used for validation consists of 20 points.

The corresponding labels for the input data are generated by a simple linear transformation. The label for each data point $x = (x_1, x_2)$ is computed as twice the value of the input features, i.e., $y = 2 \cdot x$. This linear transformation ensures that the label is directly related to the input data, which makes it easier for the model to learn the mapping. The training labels $y_{\text{train}}$ and validation labels $y_{\text{val}}$ are computed as $y_{\text{train}} = 2 \cdot x_{\text{train}}$ and $y_{\text{val}} = 2 \cdot x_{\text{val}}$, respectively.

The dataset is randomly split into training and validation datasets. The training dataset consists of 100 data points, and the validation dataset contains 20 data points. This splitting is done to ensure that the model is evaluated on unseen data, allowing for the measurement of its generalization performance.

To summarize, the input data is generated by independently sampling 2-dimensional points from a standard normal distribution, ensuring a variety of input values. The corresponding labels are generated through a simple linear scaling by a factor of 2. The dataset is split into training and validation sets, with 100 samples for training and 20 samples for validation. This dataset setup serves to evaluate the performance of a neural ODE model with a Lipschitz penalty term.

