# OpenReview forum: "Analysis of generalization capacities of Neural Ordinary Differential Equations"
_TMLR — Accepted by TMLR_

### Review · Reviewer_HgLk · 2025-08-28

**Summary Of Contributions:**

The paper establishes generalization bounds on the neural ordinary differential equations. The paper clearly extends in the prior art as it is more general (handles non-linearity in neural networks). I am not expert in this field, but I always welcome contributions in these areas, as I consider this refreshing from experimental works. Moreover, reading these papers gives me better intuition on where model shines and where are their bottlenecks.

**Audience:**

Yes

**Audience Explanation:**

I highly value theoretical works and I think that TMLR audience as well.

**Broader Impact Concerns:**

Since this is theoretical works, I do not think there are concerns.

**Claims And Evidence:**

Yes

**Claims Explanation:**

The paper is quite heavy on the math. I have not redone the proofs, as it would require substantial time investment and I might not be most suitable. But the claims seems to be well supported. What I have enjoyed is how the paper is structured. It starts with review of math for the main theorem and the corresponding proof. From this the reader is primed how the proof will be carried.

**Requested Changes:**

No changes.

---

> ### Author Response · Authors · 2025-09-19
> **Thanks**
>
> Thank you for your positive comments and appreciation of our work.

---

### Review · Reviewer_XZUr · 2025-09-02

**Summary Of Contributions:**

The paper focuses on providing a generalization analysis of Neural ODEs, an important and timely problem. However, the presentation is difficult to follow due to unclear definitions, scattered exposition, and inconsistent notations. The manuscript requires significant polishing and restructuring to improve clarity and readability.

**Audience:**

Yes

**Audience Explanation:**

Audience work on the theory of deep learning, generalization analysis, and neural differential equations would be interested in the findings.

**Claims And Evidence:**

Yes

**Claims Explanation:**

The claims are plausible and partially supported, but the evidence is neither sufficiently clear nor fully convincing due to unclear definitions and limited experiments.

**Requested Changes:**

1. The paper is difficult to follow, as the problem background and definitions are scattered across Sections 1–4 in a rather confusing manner. I suggest restructuring Section 3 to start with a clear problem definition, providing detailed explanations of the studied problem, the loss function, training data, and ERM. Afterwards, the necessary assumptions and the mathematical tools to be used should be introduced in a systematic way. If possible, it would also be very helpful to include a table summarizing the notations used throughout the paper.

2. Many notations are left undefined, which seriously affects the readability and understanding of the paper. For example, what are $A_i$ and $b_i$ in Assumption 2?  Data distribution $\mathcal{P}$ is not defined, further, is $\mathcal{P}$ and known distribution? I notice that the authors state "$R(h_\theta)$ cannot be optimized, since we do not have access to the continuous data."

3. There are many typos throughout the paper, I strongly suggest authors go through the paper carefully and polish the manuscript.

e.g.,

(1) In Lemma 3.9, $L$ should be $\ell$, and Lipschitzness of a function is not defined.

(2) Throughout the paper, $\sigma$ sometimes denote activation function, sometimes denote independent Rademacher variable.

(3) First paragraph of Section 4, $1.1$ should be $(1.1)$; $\sim y, x$ should be $(y,x)$; sometimes use $(y,x)$ sometimes use $(x,y)$, it should be consistent; in the definition of $R$, $\ell(y, h_\theta$ should be $\ell(y, h_\theta)$.

(4)  In Assumption 1, "with respect with" should be "with respect to"

(5) At the beginning of Section 5, theorem should be Theorem.

4. The paper’s lemmas and theorems do not specify which assumptions are required for their validity. While the main theorem may indeed rely on all of Assumptions 1–4, several intermediate results (e.g., Lemma 5.1) seem to depend only on a subset of them. This lack of precision is potentially misleading for readers. To improve clarity and rigor, the authors should explicitly state, for each lemma and theorem, which assumptions are being used.

5. The first explicit expression of the neural network appears only in Lemma 5.1. However, the paper presents it purely as a mathematical formula without any accompanying explanation or clarification of the underlying architecture. This lack of exposition significantly hinders readability. In addition, the dimensional consistency is problematic, as biases and weight matrices are not clearly aligned with input, hidden, and output layers. In particular, attention should be paid to the dimensional consistency of $b_i$ and the input $x\in R^d$.  A clear and systematic definition of the network architecture should be given earlier, including precise dimensions of all parameters.

6. I believe that Theorems 5.10 and 5.12 do not need to be fully stated in the main text. Instead, in the comparison section, the authors could summarize more concisely what each related work studied, what assumptions were made, what type of bound was derived, and in what sense the present results improve upon those bounds. Such a focused comparison would make the contribution much clearer. It may even be helpful to include a table that highlights the differences in assumptions, problem settings, and results between this paper and prior works.


7. The experimental section only verifies the qualitative trend that larger Lipschitz constants or higher capacity lead to larger generalization gaps. However, it does not include any direct comparison with prior works such as Marion (2023) or Bleistein $\&$ Guilloux (2023).

Overall, I am not an expert in Neural Ordinary Differential Equations, and the paper is very difficult for me to understand. This is mainly due to the lack of clarity in definitions, the scattered presentation of problem setup across multiple sections, and the inconsistent or missing explanations of notations and dimensions. These issues significantly reduce the readability of the paper and make it hard for non-experts in the area to follow the technical contributions.

---

> ### Author Response · Authors · 2025-09-19
> **Incorporated suggestions**
>
> We sincerely thank you for your thoughtful comments and suggestions, which have helped us improve the clarity and quality of the manuscript. We have carefully revised the paper in light of your feedback. Below, we provide a point-by-point response:
>
> 1. Section 3 has been restructured for better readability.
>
> 2. We have added detailed explanations of the notations used throughout the work.
>
> 3. All typographical errors have been corrected.
>
> 4. Additional details about the assumptions underlying the different lemmas are now included.
>
> 5. The inconsistencies in dimensions have been resolved, and we now provide an explicit expression for the neural network.
>
> 6. Theorems 5.10 and 5.12 have been removed, and instead we have included a comparison table to summarize the results more clearly.
> 7. Our experimental evaluation is designed to qualitatively validate the theoretical prediction that larger Lipschitz constants or higher model capacity correlate with increased generalization gaps. We do not include a direct empirical comparison with prior generalization bounds for Neural ODEs (e.g., Marion, 2023; Bleistein & Guilloux, 2024), as these results are derived in fundamentally different regimes. In particular, Marion (2023) establishes parameter-space--dependent bounds that do not involve the Lipschitz constant of the dynamical system, and thus cannot support experiments that vary this quantity. The bound given in our work is stricter in terms of $n$ for the linear case since it is $\mathcal{O}\left( \frac{1}{n^{1/2}}\right)$ for the term consisting of Lipschitz constant of weights while it is  $\mathcal{O}\left(\frac{1}{ n^{1/4}}\right)$  for the bound given in Marion (2023). Bleistein & Guilloux (2023), in contrast, obtain bounds that depend on the discretization gap of continuous-depth networks rather than the solution dynamics themselves. By contrast, our result is the first to yield a solution-space dependent bound that explicitly incorporates the Lipschitz constant of the learned dynamics. This distinction makes it possible to directly examine how generalization degrades with increasing Lipschitz constant, a type of experiment not enabled by existing bounds.

---

### Review · Reviewer_NcmR · 2025-09-16

**Summary Of Contributions:**

This paper investigates the generalization capacities of Neural Ordinary Differential Equations (Neural ODEs), focusing on deriving the first generalization bounds for both time-dependent and time-independent cases where the dynamics are governed by general nonlinear Lipschitz functions. By showing that solutions to Neural ODEs have bounded variation, the authors establish covering number and Rademacher complexity estimates, leading to explicit generalization error bounds. They further compare their results with existing works on Neural ODEs and Neural Controlled ODEs, demonstrating stricter and more informative rates. Numerical experiments on synthetic data, MNIST, and CIFAR-10 validate the theoretical insights, highlighting the role of Lipschitz continuity and parameter norms in shaping generalization performance.

**Audience:**

Yes

**Audience Explanation:**

This paper focuses on theoretical aspects of Neural ODE which is both timely and important in AI for Science.

**Broader Impact Concerns:**

I did not find any broader impact concerns because this paper is purely theoretical.

**Claims And Evidence:**

Yes

**Claims Explanation:**

1. The paper provides the first generalization bound for nonlinear Neural ODEs under Lipschitz assumptions, going beyond earlier linear analyses and establishing a new theoretical foundation.

2. The analysis is supported by empirical experiments (synthetic, MNIST, CIFAR-10), which clearly link the theoretical dependence on Lipschitz constants and parameter norms to observed generalization gaps.

**Requested Changes:**

1. Clarify the source of improvement in Section 5.1: In Section 5.1, the paper compares its bounds with Marion (2023) and Bleistein & Guilloux (2023). It would strengthen the contribution to explicitly identify the technical reason for why the new bounds achieve better rates (e.g., use of bounded variation properties, refined covering number estimates, or sharper use of Gronwall’s lemma). This would highlight the technical novelty beyond applying existing complexity arguments.

2. Discuss the tightness of the bounds: The paper could benefit from analyzing whether the derived error bounds are tight. For example, is the $O(1/\sqrt{n})$ dependence improvable under additional assumptions, or is it likely optimal given the complexity of Neural ODEs? If current literature lacks lower bounds, the authors could sketch a potential future research direction, such as investigating minimax optimality or empirical tightness under specific architectures.

3. Connect theory to applications: The results could be more impactful by discussing possible application insights. For instance, how might the derived generalization bounds guide regularization strategies in Neural ODE training (e.g., penalizing Lipschitz constants), or inform the design of ODE-based models in time-series tasks, physics-informed learning, or control problems?

---

> ### Author Response · Authors · 2025-09-19
> **Revision**
>
> Thanks for your constructive suggestions. We have revised the manuscript accordingly. Below are our detailed responses:
>
> 1. We now highlight in Remarks 4.4, 4.10  that our analysis derives a tighter covering number bound by first addressing increasing functions in $L^{1}$ and then extending to functions of bounded variation. In contrast to prior parameter-space approaches, our solution-space analysis yields bounds that depend on intrinsic trajectory properties (e.g., Lipschitz continuity) rather than parameter norms, and thus apply consistently across different parameterizations or architectures realizing the same dynamics.
>
> 2. Regarding the optimality of the bound, we agree this remains an open question due to the lack of lower-bound results in the literature. We have included a discussion outlining potential future research directions on this point.
>
> 3. To strengthen the connection to practice, we have added a new paragraph at the end of the NUMERICAL ILLUSTRATIONS Section.

---

> > ### Comment · Reviewer_NcmR · 2025-10-03
> >
> > Thanks for the response and revision. My concerns have been well addressed.

---

### Decision · Action_Editor_vc9f · 2025-10-21

**Recommendation:** Accept as is

**Audience:**

Yes

**Audience Explanation:**

As a solid contribution to the theoretical foundation (generalization properties) of neural ODEs, widely used in different areas of ML, this contribution is certainly of interest to (parts of) the TMLR audience.

**Claims And Evidence:**

Yes

**Claims Explanation:**

This submission establishes generalization bounds for non-linear ODEs under fairly general Lipschitz assumptions beyond existing linear approaches. It thereby contributes a piece to the learning theoretical foundations of different kinds of modern architectures (NODEs in this case). All reviewers appreciated this worthwhile contribution. After the author-reviewer discussion, initial concerns and questions could be clarified such that ultimately reviewers agreed that the claims made in the submission are well supported theoretically and that the empirical validation also adds to the credibility of the results.